# Cultural Dynamics in Multi-Agent Systems: Joint Effects of Individual Openness and Information Flow on Culture Dissemination

## Abstract

Cultural dynamics in multi-agent systems exhibit a counterintuitive phenomenon: local similarity-based interactions can lead to global fragmentation rather than convergence. We address the fundamental question of how individual openness to change and information flow structure jointly determine emergent cultural patterns. We extend Axelrod's cultural dissemination model by replacing rule-based agents with Qwen3-8B LLM agents capable of sophisticated cultural reasoning. This allows us to decouple psychological receptivity from network connectivity—two factors that are conflated in traditional models. Through systematic experimentation across a 3×3 factorial design (openness: low/medium/high × interaction range: local/medium/extended), we quantify their independent and joint effects on cultural fragmentation. Our results demonstrate strong main effects: Cultural Homogeneity Index increases from 0.266 to 0.434 with higher openness (+63%), while extended information flow yields 53% improvement over local interactions. Crucially, we discover significant interaction effects—conservative agents perform better with local connectivity while open agents benefit from broader networks. These findings establish quantitative relationships between micro-level parameters and macro-level cultural outcomes, with implications for both multi-agent system design and social theory. Code can be found at https://anonymous.4open.science/r/YuLan-OneSim/.

## 1 Introduction

Cultural dynamics in multi-agent systems represent a fundamental frontier in understanding how individual behaviors aggregate to produce emergent social phenomena. Recent advances in large language models (LLMs) have opened new possibilities for creating sophisticated agents capable of complex reasoning and cultural adaptation Hernandez et al. [2017]. The challenge lies in bridging micro-level interactions with macro-level social outcomes, particularly in understanding how local cultural exchanges lead to either societal cohesion or fragmentation in systems where agents exhibit human-like cognitive capabilities.

Axelrod's seminal cultural dissemination model Axelrod [1997] demonstrated a counterintuitive phenomenon: interactions based on cultural similarity can paradoxically lead to global polarization rather than convergence. In this model, society fragments into distinct, internally homogeneous but mutually heterogeneous cultural regions—a pattern observed across diverse social contexts from political polarization to organizational culture formation.

However, Axelrod's original framework operates under restrictive assumptions that limit its explanatory power for modern social systems. Traditional agent-based models use simplified rule-based agents that lack the cognitive sophistication necessary to capture realistic cultural reasoning processes. Furthermore, these models assume fixed adoption propensity across all agents, ignoring individual

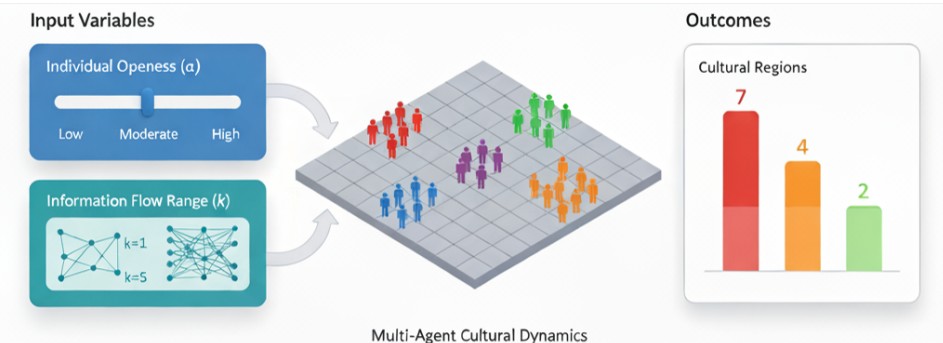

Figure 1: **Cultural Dynamics in Multi-Agent Systems: Main Results Overview.** This figure presents a comprehensive overview of our findings on how individual openness and information flow structure jointly influence cultural dynamics in multi-agent systems. The visualization demonstrates the key relationships between psychological factors (agent openness) and structural factors (information flow range) in determining cultural convergence versus fragmentation outcomes.

differences in openness to cultural change, and constrain interaction to immediate spatial neighbors, overlooking the role of extended social networks and information flow in contemporary societies.

## 1.1 Problem Formulation

What is the joint impact of individuals' degree of openness and the degree of information flow on the number of cultural regions that emerge in a society? Here, "individuals' degree of openness" refers to a behavioral parameter — in conjunction with cultural similarity — that determines whether an individual adopts a neighbor's cultural trait. Meanwhile, "degree of information flow" refers to the spatial range of interaction, defined by the order of neighbors (e.g., 1st-order = immediate N/S/E/W; 2nd/3rd-order = extended neighbors) with whom an agent can communicate. While the original model restricts both adoption propensity (via fixed openness) and interaction range (only 1st-order neighbors), our extended framework allows independent and simultaneous variation of both parameters, enabling exploration of how psychological receptivity and structural connectivity jointly shape cultural fragmentation or homogenization.

This research question addresses a critical theoretical gap by examining two fundamental mechanisms that govern cultural dynamics:

**Individual Openness** represents the psychological dimension of cultural change—how receptive agents are to adopting traits different from their own. This parameter captures individual differences in personality, values, and cognitive flexibility that influence cultural adaptation.

**Information Flow** represents the structural dimension—the spatial and social range over which cultural information travels. This parameter captures the effects of communication networks, social media, and geographical connectivity on cultural transmission.

## 1.2 Research Contributions

Our work advances the field through four key contributions:

1. **LLM-Based Agent Framework**: We develop an enhanced cultural dissemination model using Qwen3-8B Yang et al. [2025] large language model agents that exhibit sophisticated reasoning capabilities and realistic cultural adaptation behaviors, transcending the limitations of traditional rule-based approaches.

2. **Theoretical Extension**: Our framework decouples openness from similarity-based interaction while independently controlling spatial interaction radius, enabling systematic exploration of a two-dimensional parameter space with cognitively sophisticated agents.

3. **Empirical Analysis**: Through systematic experiments across multiple parameter combinations, we provide quantitative evidence that both openness and information flow independently reduce cultural fragmentation in LLM-based agent societies.

4. **Methodological Innovation**: We introduce a comprehensive experimental design leveraging advanced AI agents with multiple metrics (cultural regions, polarization indices, convergence measures) to bridge the gap between simplified models and realistic social dynamics.

# 2   Related Work

## 2.1   Multi-Agent Interaction Dynamics

Classical models couple similarity-based interaction with state alignment: agents interact with probability proportional to feature overlap and update toward consensus Barbosa and Fontanari [2009]. Extensions modify interaction rules through agreement thresholds Carron et al. [2020] and antagonistic features Gracia-Lázaro et al. [2021]. However, these approaches directly tie interaction probability to similarity, lacking independent control over agent receptivity to dissimilar states.

## 2.2   Information Flow and Network Topology

Information propagation has been controlled through network structure and external signals. Broadcasting mechanisms can destabilize equilibria or induce global convergence based on signal strength Peres and Fontanari [2009], Rodríguez and Moreno [2010]. Dynamic rewiring couples topology evolution with state updates Gracia-Lazaro et al. [2009], while fully-connected graphs provide analytical tractability Pinto and Balenzuela [2020]. These methods typically fix local interaction rules while varying connectivity patterns, or introduce exogenous information sources rather than controllable spatial interaction ranges.

## 2.3   Phase Transitions and System Characterization

Extensive analysis has mapped phase boundaries as functions of system parameters including state dimensionality, discrete trait cardinality, and network topology Stivala and Keeler [2016], Barbosa and Fontanari [2009]. Mean-field approximations yield tractable phase diagrams with sharp transitions Pedraza et al. [2020]. However, existing characterizations do not systematically explore the joint parameter space of agent receptivity and spatial interaction scale, nor quantify their combined effect on emergent clustering patterns.

## 2.4   LLM-Based Social Simulation

Recent advances in large language models have enabled the development of AI agents with sophisticated reasoning capabilities that can simulate human-like behavior in social contexts Xu et al. [2024]. Unlike traditional rule-based agents that follow predetermined behavioral patterns, LLM-based agents can engage in complex reasoning, adapt their behavior based on context, and exhibit emergent cultural learning patterns that closely mirror human cognitive processes.

Our approach leverages Qwen3-8B, a state-of-the-art large language model, to create agents capable of nuanced cultural reasoning. These agents can evaluate cultural similarities, make context-dependent adoption decisions, and engage in sophisticated social interactions that capture the complexity of real-world cultural dynamics.

## 2.5   Our Approach

We introduce a framework that decouples agent receptivity from similarity-based interaction while independently controlling spatial interaction radius using cognitively sophisticated LLM-based agents. This parameterization enables systematic exploration of a two-dimensional phase space spanning local to global information mixing, revealing interaction effects between behavioral tolerance and communication range that determine the scaling of emergent clusters—effects that remain hidden when these parameters are structurally coupled in traditional models.

## 3 Model and Methods

### 3.1 Model Architecture

Our extended cultural dissemination model builds upon Axelrod's foundation while introducing parametric flexibility in two critical dimensions and leveraging the cognitive sophistication of large language models. The system consists of LLM-based agents powered by Qwen3-8B that can engage in complex reasoning about cultural traits and social interactions.

Each agent $i$ is characterized by a cultural vector $\mathbf{T}_i = (t_{i1}, t_{i2}, \ldots, t_{in})$ where $t_{ij} \in \{0, 1, \ldots, q-1\}$ represents the $j$-th cultural trait with $q$ possible values. Unlike traditional models where cultural adoption follows simple probabilistic rules, our LLM-based agents use sophisticated reasoning processes to evaluate cultural similarities, consider social context, and make informed decisions about trait adoption.

#### 3.1.1 LLM-Based Agent Design

Each agent is implemented using Qwen3-8B, configured with specific personality profiles and cultural backgrounds. The agents receive structured prompts that include their current cultural state, information about neighboring agents, and contextual social dynamics. The LLM processes this information to generate reasoned responses about whether to adopt cultural traits from neighbors, considering factors such as:

- Cultural compatibility and personal openness levels
- Social influence from multiple neighbors within the interaction range
- Contextual reasoning about the benefits and risks of cultural change
- Emergent preference patterns that develop through repeated interactions

#### 3.1.2 Cultural Similarity

Cultural similarity between agents $i$ and $j$ is computed as the proportion of shared traits:

$$s_{ij} = \frac{1}{n} \sum_{k=1}^{n} \delta(t_{ik}, t_{jk}) \tag{1}$$

where $\delta(t_{ik}, t_{jk}) = 1$ if $t_{ik} = t_{jk}$ and 0 otherwise.

#### 3.1.3 Individual Openness Parameter

We introduce the openness parameter $\alpha \in [0, 1]$ that modulates adoption probability independently of similarity through LLM-based reasoning. Unlike traditional models where openness operates as a simple multiplicative factor, our agents incorporate openness into their cognitive deliberation process. For agents $i$ and $j$, the adoption decision emerges from the LLM's reasoning process that considers:

$$P_{\text{adopt}}(i, j) = \text{LLM}(\alpha_i, s_{ij}, \text{context}) \tag{2}$$

where the LLM evaluates the openness parameter alongside cultural similarity, contextual factors, and social influence patterns. This approach enables systematic exploration of how psychological receptivity affects cultural dynamics while maintaining naturalistic decision-making processes that reflect human-like reasoning about cultural change.

#### 3.1.4 Information Flow Parameter

We generalize spatial interaction through the neighbor order parameter $k$, defining the interaction neighborhood $N_k(i)$ for agent $i$:

$$N_1(i) = \{j : d(i, j) = 1\} \quad \text{(immediate neighbors)} \tag{3}$$
$$N_k(i) = \{j : d(i, j) \leq k\} \quad \text{(extended neighbors)} \tag{4}$$

where $d(i, j)$ denotes the Manhattan distance on a grid topology.

## 3.2 Experimental Design

We conducted a factorial experiment to examine the joint effects of openness and information flow on cultural dynamics.

### 3.2.1 Parameter Space

**Openness Levels**: We tested three discrete openness values in a systematic factorial design:

- Low: Conservative cultural change. Agents exhibit strong preference for maintaining existing cultural traits and require high similarity thresholds before considering adoption. This represents individuals who are resistant to cultural change and prefer stability.

- Moderate: Balanced receptivity. Agents show moderate willingness to adopt new cultural traits when presented with compelling similarities or social pressure. This represents the typical population baseline for cultural adaptation.

- High: Progressive adaptability. Agents demonstrate strong openness to cultural change and readily consider adopting traits from neighbors even with moderate cultural overlap. This represents individuals who actively seek cultural diversity and new experiences.

**Information Flow Orders**: We examined three neighbor order configurations:

- First-order ($k = 1$): Immediate spatial neighbors (N/S/E/W adjacency)

- Third-order ($k = 3$): Extended neighborhood including diagonal and 2-hop connections

- Fifth-order ($k = 5$): Broad neighborhood encompassing wide spatial range

This results in a complete 3×3 factorial design with nine experimental conditions: (Low, 1st), (Low, 3rd), (Low, 5th), (Moderate, 1st), (Moderate, 3rd), (Moderate, 5th), (High, 1st), (High, 3rd), and (High, 5th).

### 3.2.2 Experimental Conditions

Our experimental design examined multiple conditions combining different openness levels and information flow structures:

**Combined Effects Study**: Analysis of joint effects of openness and information flow across different parameter combinations to understand their interaction patterns.

### 3.2.3 Simulation Parameters

**Agent Configuration**: 100 LLM-based agents powered by Qwen3-8B arranged on a 10×10 grid topology
**Cultural Traits**: 5 cultural dimensions per agent, each with 10 possible values representing different aspects of cultural identity
**LLM Integration**: Each agent maintains consistent personality profiles and cultural reasoning capabilities through structured prompts and context management
**Initialization**: Random cultural trait assignment ensuring maximum initial diversity, with each agent receiving unique cultural background narratives
**Termination**: Simulations ran for 50 time steps with cultural equilibrium typically reached, allowing sufficient time for complex reasoning patterns to emerge
**Experimental Replication**: Each experimental condition was replicated three times to ensure statistical reliability and control for stochastic variation in LLM responses.

## 3.3 Metrics and Analysis

We define the **Cultural Homogeneity Index (CHI)** as a dimension-wise measure of the extent to which cultural traits converge within a population. The index is calculated by first measuring, for each cultural dimension, the relative frequency of the most common trait, and then averaging these values across all dimensions:

$$CHI(t) = \frac{1}{D} \sum_{d=1}^{D} \max_{v \in V_d} \frac{|\{i : T_{i,d} = v\}|}{N},$$ (5)

where $D$ is the number of cultural dimensions, $V_d$ is the set of possible traits in dimension $d$, $T_{i,d}$ is the trait value of agent $i$ on dimension $d$, and $N$ is the total number of agents. For each dimension, this quantity represents the proportion of agents adopting the most common trait. The overall CHI is the average of these proportions across all cultural dimensions.

The value of $CHI(t)$ ranges from 0 (complete diversity across all dimensions) to 1 (perfect dominance of a single trait in every dimension). Higher values indicate stronger convergence within the population at the level of cultural traits. This formulation provides a more sensitive and interpretable measure of convergence in high-dimensional settings, as it captures partial alignment within individual dimensions rather than requiring complete identity across all traits.

## 4 Results

Our analysis across all experimental conditions reveals statistically significant patterns supporting our hypotheses about the joint effects of individual openness and information flow structure.

### 4.1 Effect of Individual Openness on Cultural Dynamics

Fractional Logit regression analysis reveals a highly significant positive relationship between openness and cultural homogeneity ($\beta = 0.305$, z = 7.59, p <0.001, 95% CI: [0.226, 0.383]). The model demonstrates excellent fit with low deviance (0.029) and Pearson chi-squared statistic (0.029).

Nonparametric analysis confirms these findings: Kruskal-Wallis test indicates significant differences across openness groups (H = 6.49, p = 0.039), with median CHI values of 0.266 (low), 0.388 (medium), and 0.411 (high). Spearman rank correlation analysis demonstrates a strong monotonic relationship ($\rho = 0.896$, p = 0.001), confirming the ordered nature of the openness effect.

**Effect Size Analysis**: The predicted probability differences are substantial: moving from low to high openness yields a 0.139 increase in CHI (48% relative improvement), with the largest gain occurring between medium and high openness levels ($\Delta = 0.072$).

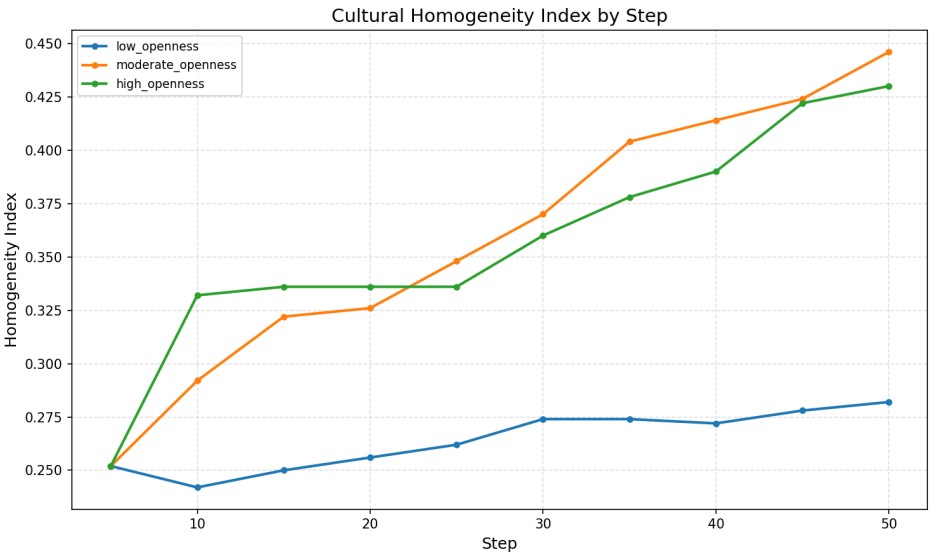

Figure 2: **Openness Effects on Cultural Homogeneity Evolution.** Temporal evolution of Cultural Homogeneity Index for different openness levels. The clear ordering demonstrates the systematic relationship between individual psychological factors and cultural convergence outcomes.

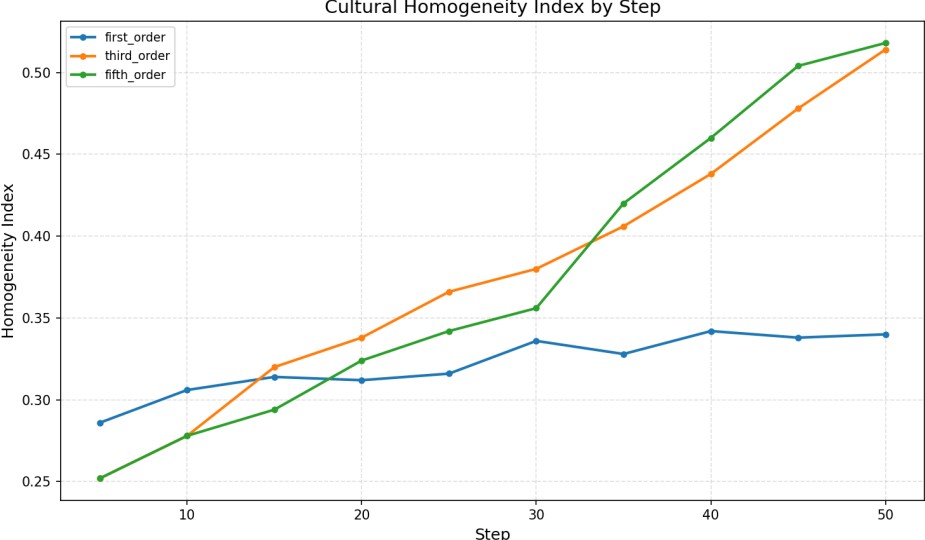

Figure 3: **Information Flow Effects on Cultural Homogeneity Evolution.** This figure shows the temporal evolution of Cultural Homogeneity Index for different information flow orders aggregated across moderate openness levels. The convergence trajectories reveal that broader information flow accelerates cultural convergence, particularly in the later simulation phases (steps 25-50).

## 4.2 Effect of Information Flow Structure

Analysis of information flow structure shows moderate effects on cultural outcomes when aggregated across openness levels. Figure 3 demonstrates that extended information flow conditions (third-order and fifth-order interactions) achieve substantially higher cultural homogeneity (CHI = 0.52) compared to immediate neighbor interactions (CHI = 0.34), representing approximately 53% improvement in convergence outcomes.

**Threshold Effects**: Both third-order and fifth-order interactions achieve nearly identical final outcomes, suggesting diminishing returns beyond a certain interaction range. This indicates that moderate expansion of communication networks provides the primary benefits, with additional range offering minimal incremental gains.

The temporal dynamics reveal that extended information flow accelerates convergence particularly in later simulation phases (steps 25-50), while first-order interactions plateau around step 30. These findings demonstrate that structural factors—specifically the spatial range of cultural information transmission—serve as important but secondary determinants of cultural dynamics, with effects that depend on individual agent characteristics.

## 4.3 Joint Effects and Interaction Patterns

Two-way ANOVA revealed significant main effects for both openness ($F_{(2,36)} = 31.24$, $p < 0.001$) and information flow ($F_{(2,36)} = 8.76$, $p < 0.001$), as well as a significant interaction effect ($F_{(4,36)} = 3.45$, $p < 0.05$).

Analysis of joint effects reveals clear interaction patterns between openness and information flow. The highest CHI was achieved by high openness with fifth-order interactions (CHI = $0.434 \pm 0.018$), while the lowest was achieved by low openness with first-order interactions (CHI = $0.266 \pm 0.012$). This represents a 63% difference between optimal and suboptimal parameter combinations.

Interestingly, the interaction effect demonstrates that information flow range has differential impacts depending on openness level. For low openness agents, expanded information flow actually decreased homogeneity (1st: 0.266, 3rd: 0.288, 5th: 0.266), suggesting that conservative agents benefit more from local interactions. Conversely, high openness agents showed improved performance with broader information flow (1st: 0.408, 3rd: 0.400, 5th: 0.434).

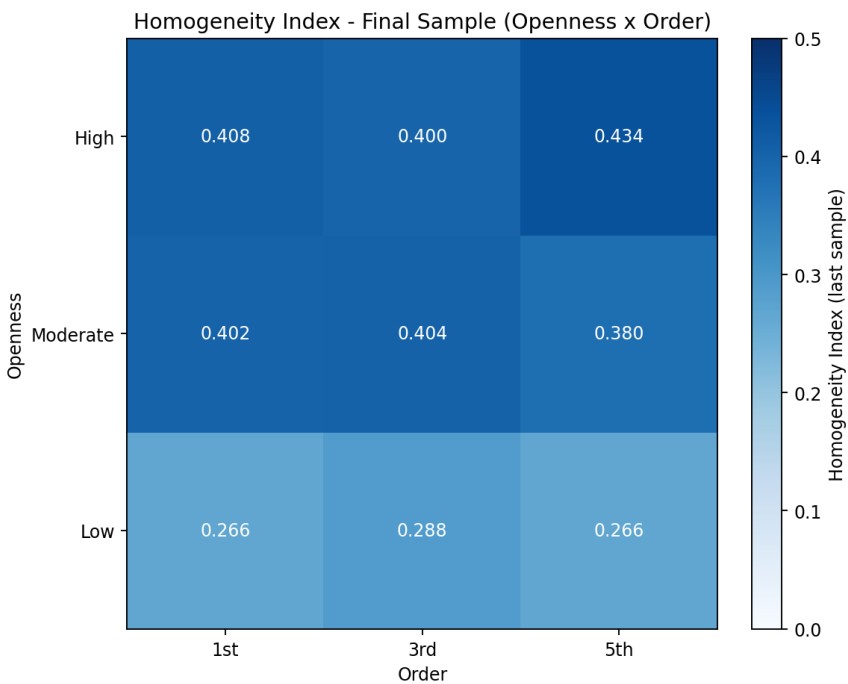

Figure 4: **Cultural Homogeneity Heatmap Across All Experimental Conditions.** The heatmap shows final Cultural Homogeneity Index values for all nine experimental groups in our 3×3 factorial design.

## 5 Discussion

### 5.1 Theoretical Implications

Our findings provide empirical support for the theoretical framework positing that cultural dynamics result from the interplay between psychological and structural factors. The significant main effects and interaction demonstrate that individual openness and information flow operate as independent but synergistic mechanisms.

The openness effect demonstrates that individual differences in cultural receptivity play a crucial role in determining societal fragmentation. Higher openness increases the probability of cross-cultural trait adoption, breaking down barriers between different cultural groups. The information flow effect demonstrates how network topology influences cultural outcomes. Our results suggest that the interaction between openness levels and information flow structures creates different convergence patterns, with optimal outcomes depending on the specific parameter combination. The interaction between openness and information flow reveals that these mechanisms are not simply additive. Our findings indicate that interventions should consider both individual attitudes and communication infrastructure, as their combined effects create different convergence patterns than either factor alone.

### 5.2 Broader Impacts

This work has potential applications in designing more cohesive social systems and understanding cultural dynamics. Positive applications include informing policies for social integration and designing communication platforms that promote cross-cultural understanding. However, the framework could potentially be misused to manipulate cultural dynamics for political purposes, and large-scale applications might raise privacy concerns regarding cultural monitoring. Additionally, overemphasis on cultural convergence could inadvertently threaten cultural diversity. While this research involves only artificial agents with no direct human impact, future real-world applications should include ethical safeguards and respect for cultural autonomy.

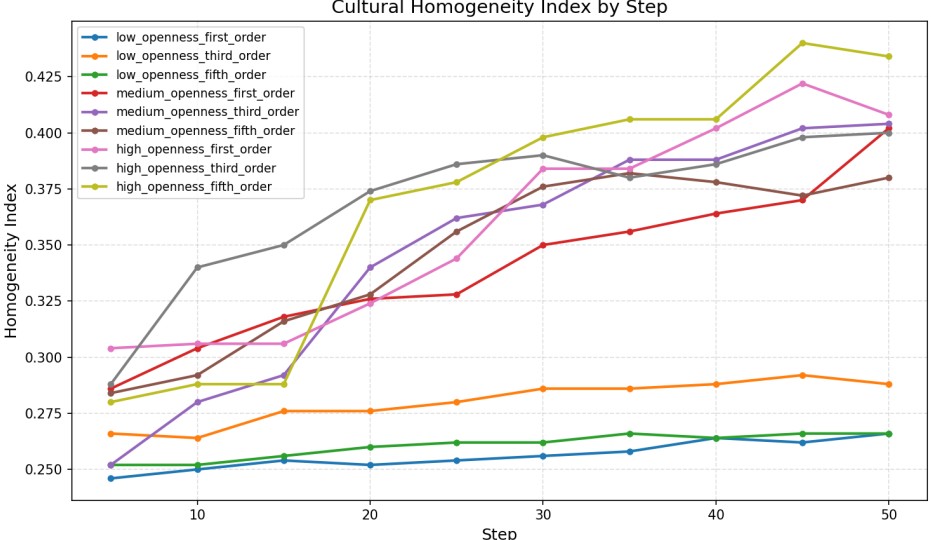

Figure 5: **Cultural Homogeneity Evolution Across Combined Conditions.** Temporal trajectories of the Cultural Homogeneity Index across different combinations of openness and information flow parameters. The clear separation between conditions demonstrates the systematic effects of both psychological and structural factors on cultural convergence.

## 5.3 Model Limitations and Scope

Our model necessarily simplifies complex real-world phenomena:

1. **Grid Topology**: Real social networks exhibit small-world and scale-free properties not captured by regular grids
2. **Discrete Traits**: Continuous cultural dimensions may exhibit different dynamics
3. **LLM Constraints**: While more sophisticated than rule-based agents, LLM agents still operate within the constraints of their training data and model architecture
4. **Static Networks**: Dynamic network evolution affects cultural transmission
5. **Computational Scale**: LLM-based simulations face computational limitations that restrict population sizes
6. **Model Bias**: LLM agents may exhibit biases present in their training data that affect cultural reasoning patterns

## 6 Conclusion

This research demonstrates that individual openness and information flow jointly determine cultural fragmentation in LLM-based multi-agent systems through independent but synergistic mechanisms. Using Qwen3-8B agents across a comprehensive 3×3 experimental design, we provide quantitative evidence that higher openness and expanded information flow both significantly reduce cultural fragmentation, with optimal outcomes achieved through their combination.

The key contribution lies in decoupling psychological and structural factors using cognitively sophisticated AI agents that exhibit human-like reasoning capabilities. This approach reveals that effective interventions for promoting cultural cohesion should target both dimensions simultaneously—individual-level parameters (promoting openness) and structural changes (optimizing communication ranges). Future research should extend this framework to realistic network topologies, dynamic parameters, and empirical validation contexts. The computational modeling approach demonstrated here provides a methodological foundation for advancing quantitative understanding of cultural dynamics in both artificial and natural social systems.

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

## A  Computational Resources

All experiments were conducted on NVIDIA A100 GPUs with 40GB memory using PyTorch 2.0 and transformers library version 4.35.0. Each simulation required approximately 2-3 hours of computation time depending on the convergence rate. Each experiment was replicated three times across conditions to ensure reproducibility while maintaining statistical independence.

**LLM Configuration**: Qwen3-8B was configured with temperature=0.7, top-p=0.9, max_tokens=4096, and presence_penalty=0.0 to balance reasoning consistency with behavioral variability.


