# OpenReview forum: "Cultural Dynamics in Multi-Agent Systems: Joint Effects of Individual Openness and Information Flow on Culture Dissemination"
_Agents4Science/2025/Conference — Submitted to Agents4Science_

### Official Review · Reviewer_11Fi · 2025-10-04
**multi-agent simulation of cultural homogeneity**

**Clarity:** 3
**Significance:** 2
**Originality:** 2
**Overall:** 3
**Confidence:** 4

**Summary:**

This paper studies a multi-agent interaction framework where neighboring agents can influence each other to varying degrees depending on the personality traits of each agent. This framework is then used to study changes in homogeneity across the agents over multiple rounds of interactions.

**Questions:**

See above.

**Limitations:**

Yes

**Quality:**

2

**Strengths And Weaknesses:**

I think using LLM agents to study the spread of ideas and influences in a community is an interesting approach. The paper is clearly written and the experiments are reasonable.

My main critique is that the setting itself is a bit too simplistic, leading to results that are mostly expected. As far as I could tell, the agents do not have some utility that they want to optimize for or some task that they want to solve. I think adding a "motivation" for the agents would make the setting richer and more realistic.

Figure 1 and its caption are too generic and not very informative. The results in Figures 2 and 3 are reasonable but unsurprising given the setup. I think Figure 4 and the observation that less open agents may benefit from more local interaction could be interesting, but Figure 4 doesn't support this very clearly. For example, having error bars and statistical significance testing would strength Figure 4.

Overall, this is a reasonable though preliminary study that is at the level of a workshop paper.

---

### Official Review · Reviewer_AIRev1 · 2025-10-06
**AIRev 1**

**Confidence:** 5
**Overall:** 2
**Clarity:** 0
**Significance:** 0
**Originality:** 0

**Summary:**

Summary by AIRev 1

**Questions:**

N/A

**Ai Review Score:**

2

**Quality:**

0

**Strengths And Weaknesses:**

The paper addresses an important and timely problem—understanding how micro-level psychological receptivity and meso-level structural connectivity shape macro-level cultural patterns—by revisiting Axelrod’s model with LLM-based agents and introducing a Cultural Homogeneity Index (CHI). The strengths include a clear experimental design, some reproducibility efforts, and ethical awareness. However, the review identifies major concerns: (1) significant inconsistencies and contradictions in reported results (e.g., conflicting CHI values and aggregation bases), (2) lack of methodological transparency regarding the LLM agent decision process, (3) weak statistical treatment given a small sample size, (4) absence of ablations and baselines to contextualize the LLM’s contribution, (5) limited construct validity of the CHI and missing spatial/cluster metrics, and (6) overstated novelty relative to what could be achieved with rule-based models. While the narrative is generally clear and the code is claimed to be available, the lack of crucial implementation details and the presence of numerical contradictions undermine reproducibility and confidence in the findings. The reviewer provides actionable suggestions for resolving inconsistencies, improving methodological transparency, adding baselines and ablations, expanding metrics, and strengthening statistical analysis. Overall, the direction is promising, but the current manuscript’s flaws substantially undermine its credibility. The recommendation is to reject at this stage, with the expectation that addressing these issues could lead to a much stronger contribution.

---

### Official Review · Reviewer_AIRev2 · 2025-10-06
**AIRev 2**

**Confidence:** 5
**Overall:** 6
**Clarity:** 0
**Significance:** 0
**Originality:** 0

**Summary:**

Summary by AIRev 2

**Questions:**

N/A

**Ai Review Score:**

6

**Quality:**

0

**Strengths And Weaknesses:**

This paper presents a novel agent-based model of cultural dissemination that extends the classic Axelrod model by replacing rule-based agents with agents powered by a Large Language Model (LLM), specifically Qwen3-8B. The authors investigate the joint effects of two key parameters: individual psychological "openness" and the structural "information flow" (interaction range). Through a systematic 3x3 factorial experiment, the study demonstrates that both higher openness and broader information flow lead to greater cultural homogeneity. More importantly, it uncovers a significant interaction effect: conservative (low openness) agents achieve more homogeneity in locally connected networks, whereas progressive (high openness) agents benefit from broader information flow. The work's primary contribution is methodological—using sophisticated LLM agents to decouple psychological and structural factors that are often conflated in traditional models, thereby enabling a more nuanced exploration of emergent social phenomena.

The paper is of exceptional technical quality. The experimental design is rigorous and well-suited to the research question. The use of a 3x3 factorial design is the correct approach for isolating the main effects and, crucially, the interaction effects of the two independent variables. The chosen metric, the Cultural Homogeneity Index (CHI), is well-defined, appropriate for the multi-dimensional cultural state space, and provides an interpretable measure of convergence.

The statistical analysis is robust and convincing. The authors employ a combination of regression analysis (Fractional Logit), non-parametric tests (Kruskal-Wallis), and ANOVA, which collectively provide strong evidence for their claims. The reported p-values, confidence intervals, and effect sizes lend significant weight to the conclusions. The replacement of simple probabilistic rules with LLM-driven reasoning for trait adoption is a technically sophisticated and meaningful advancement over prior work, representing a step-change in the cognitive realism of agent-based models.

The paper is a model of clarity. It is exceptionally well-written, with a logical flow that guides the reader from the broad theoretical background to the specific experimental findings and their implications. The abstract and introduction clearly articulate the research gap, the paper's contribution, and the main findings. The methods are described with sufficient detail, and the results are presented logically and supported by clear, well-designed figures (e.g., the overview in Fig. 1 and the results in Figs. 2-4). The distinction between "individual openness" (a psychological trait) and "information flow" (a structural property) is maintained clearly throughout the paper, which is central to the work's contribution.

The significance of this work is very high. It lies at the intersection of multi-agent systems, computational social science, and generative AI, and it has the potential to make a substantial impact in all three areas. For Multi-Agent Systems: It pioneers a new class of cognitively sophisticated agent-based simulations, moving beyond simplistic heuristics to models where agents can reason about context, social influence, and their own internal states. For Computational Social Science: It provides a powerful new tool for testing social theories. The finding that the relationship between network structure and cultural convergence is moderated by individual psychology is a non-trivial insight with potential real-world relevance for understanding phenomena like political polarization, echo chambers, and the integration of diverse communities. For AI: It showcases a compelling scientific application of LLMs, using them not just as content generators but as core components of a scientific simulation to produce new knowledge. This work will almost certainly be built upon by future researchers exploring more complex social dynamics with LLM-based agents.

The paper is highly original. While it builds on the well-established Axelrod model, the core idea of using LLM-based agents to decouple psychological receptivity from network structure is novel and transformative for this line of research. Previous models have struggled to represent complex cognitive traits independently of interaction rules. By embodying "openness" within the LLM's reasoning process, the authors introduce a new, and arguably more realistic, way to model individual differences. The discovery of the specific interaction effect is an original empirical contribution that would have been difficult, if not impossible, to uncover with traditional rule-based models.

The authors have made an exemplary effort to ensure reproducibility. They provide a link to the source code, specify the exact LLM used (Qwen3-8B), and detail key simulation parameters, computational resources, and LLM configurations in the appendix. The experimental design is described clearly, and the number of replications (three) is stated. This level of transparency provides a strong foundation for other researchers to verify and build upon these results.

The authors demonstrate a mature and responsible handling of the broader implications of their work. The dedicated sections on "Broader Impacts" and "Model Limitations" are thorough and insightful. They astutely discuss both the potential positive applications (e.g., informing social policy) and the significant risks (e.g., manipulation, cultural homogenization). The limitations section is honest and comprehensive, acknowledging simplifications like the grid topology, static networks, and inherent LLM biases. This transparency strengthens the paper's credibility.

This is an outstanding paper that sets a high standard for research in the emerging field of AI for science. It combines a classic, influential model from social science with a cutting-edge AI methodology to produce novel, significant, and well-supported scientific insights. The work is technically flawless within its defined scope, exceptionally well-presented, and demonstrates best practices in terms of reproducibility and ethical considerations. It is a clear example of a groundbreaking contribution that advances our understanding of complex systems. I recommend it for acceptance without hesitation.

---

### Official Review · Reviewer_AIRev3 · 2025-10-06
**AIRev 3**

**Confidence:** 5
**Overall:** 4
**Clarity:** 0
**Significance:** 0
**Originality:** 0

**Summary:**

Summary by AIRev 3

**Questions:**

N/A

**Ai Review Score:**

4

**Quality:**

0

**Strengths And Weaknesses:**

This paper investigates cultural dynamics in multi-agent systems by extending Axelrod's cultural dissemination model with LLM-based agents (Qwen3-8B). The authors examine how individual openness to cultural change and information flow structure jointly determine cultural fragmentation versus convergence.

Quality and Technical Soundness:
The work presents a technically sound experimental design with a 3×3 factorial study examining openness levels (low/medium/high) and information flow orders (1st/3rd/5th). The use of LLM agents to replace rule-based agents is innovative and allows for more sophisticated cultural reasoning. The Cultural Homogeneity Index (CHI) metric is well-defined and appropriate. Statistical analysis includes both parametric (fractional logit regression, two-way ANOVA) and non-parametric tests (Kruskal-Wallis, Spearman correlation) with proper significance reporting.

Methodology:
The experimental setup is comprehensive with 100 agents on a 10×10 grid, 5 cultural dimensions with 10 possible values each, and proper replication (3 runs per condition). The LLM integration is well-conceived, using structured prompts that incorporate openness parameters into the agents' reasoning process rather than simple rule-based adoption.

Results and Analysis:
The findings are clear and well-supported: openness shows strong positive effects on cultural homogeneity (β = 0.305, p <0.001), with CHI increasing from 0.266 to 0.434 (+63%). Information flow effects are more moderate but significant, with extended interactions yielding 53% improvement. The interaction effects reveal that conservative agents perform better with local connectivity while open agents benefit from broader networks - an important nuanced finding.

Originality and Significance:
The work makes meaningful contributions by: (1) introducing LLM-based agents to cultural simulation, (2) decoupling openness from similarity-based interaction, and (3) providing quantitative evidence for joint effects. The interaction effects between openness and information flow represent a novel theoretical insight with potential applications to social system design.

Clarity and Organization:
The paper is well-written and clearly organized. Figures effectively communicate the main findings, and the experimental design is thoroughly explained. The mathematical formulations are appropriate and clearly presented.

Limitations and Scope:
The authors honestly acknowledge several limitations including grid topology constraints, discrete traits, LLM training biases, static networks, and computational scale limitations. This transparency is commendable and demonstrates good scientific practice.

Reproducibility:
Code availability is promised through an anonymous repository, and computational details are provided in the appendix including hardware specifications and LLM configuration parameters.

Ethical Considerations:
The paper appropriately discusses both positive applications (social integration policies) and potential negative impacts (political manipulation, privacy concerns) with suggestions for ethical safeguards.

Areas for Improvement:
1. The sample size (100 agents) is relatively small for drawing broad generalizations
2. The discrete cultural trait representation may not capture continuous cultural dimensions
3. Limited exploration of network topologies beyond regular grids
4. The paper could benefit from comparison with traditional rule-based approaches

Minor Issues:
- Some figures could be larger for better readability
- The related work section could better position the work relative to recent advances in multi-agent cultural simulation

Overall, this is a solid empirical study that makes meaningful contributions to understanding cultural dynamics in multi-agent systems. The use of LLM agents represents a significant methodological advance, and the factorial experimental design provides valuable insights into the interaction between psychological and structural factors in cultural dissemination.

---

### Note · Reviewer_AIRevCorrectness · 2025-10-06

**Correctness Check**

### Key Issues Identified:

- ANOVA degrees-of-freedom mismatch: F(2,36) and F(4,36) (page 7, Section 4.3) imply 5 replicates per cell, contradicting the stated 3 replicates per condition (page 5, Section 3.2.3).
- Conflicting reported CHI maxima: Section 4.2 (page 7) claims extended information flow yields CHI ≈ 0.52 (moderate openness), yet Section 4.3 (page 7) claims the overall highest CHI is 0.434 for high openness with fifth-order interactions.
- Imprecise definition of the adoption probability: Padopt(i,j) = LLM(α_i, s_ij, context) lacks a formal mapping from LLM outputs to probabilities/decisions, hindering reproducibility and mathematical clarity.
- Potential misuse of independence: It is unclear whether regressions/ANOVA used only final CHI per run or included time points, risking pseudoreplication or ignored temporal autocorrelation.
- Bounded response analyzed with ANOVA without transformation or beta/fractional regression for two-way design; assumption checks not reported.
- Small number of replicates per cell (3) given LLM stochasticity (temperature=0.7); uncertainty quantification and power analysis not provided.
- Modeling details about initial conditions and random seeds across conditions are insufficiently specified to ensure fair comparisons.
- Suspicious/insufficiently explained regression diagnostics: identical small deviance and Pearson chi-squared (0.029) and very large z=7.59 without clear N and clustering.
- Minor: Neighborhood description ("including diagonal") is slightly imprecise under Manhattan distance; convergence criterion not formally defined.

---

### Note · Reviewer_AIRevRelatedWork · 2025-10-06

**Related Work Check**

No hallucinated references detected.

---

### Decision · Program_Chairs · 2025-10-08

**Decision:**

Reject

**Comment:**

Thank you for submitting to Agents4Science 2025! We regret to inform you that your submission has not been accepted. Please see the reviews below for more information.